

# Association between the number and size of intrapulmonary lymph nodes and chronic obstructive pulmonary disease severity

Anton Schreuder[1], Colin Jacobs[1,2], Ernst T. Scholten[1], Mathias Prokop[1], Bram van Ginneken[1,2], David A. Lynch[3], Cornelia M. Schaefer-Prokop[1,4] and COPDGene Investigators

[1] Department of Radiology and Nuclear Medicine, Radboudumc, Nijmegen, The Netherlands
[2] Fraunhofer MEVIS, Bremen, Germany
[3] Department of Radiology, National Jewish Medical and Research Center, Denver, CO, United States of America
[4] Department of Radiology, Meander Medisch Centrum, Amersfoort, The Netherlands

Corresponding author
Anton Schreuder,
antoniusschreuder@gmail.com

## ABSTRACT

**Purpose**. One of the main pathophysiological mechanisms of chronic obstructive pulmonary disease is inflammation, which has been associated with lymphadenopathy. Intrapulmonary lymph nodes can be identified on CT as perifissural nodules (PFN). We investigated the association between the number and size of PFNs and measures of COPD severity.

**Materials and Methods**. CT images were obtained from COPDGene. 50 subjects were randomly selected per GOLD stage (0 to 4), GOLD-unclassified, and never-smoker groups and allocated to either "Healthy," "Mild," or "Moderate/severe" groups. 26/350 (7.4%) subjects had missing images and were excluded. Supported by computer-aided detection, a trained researcher prelocated non-calcified opacities larger than 3 mm in diameter. Included lung opacities were classified independently by two radiologists as either "PFN," "not a PFN," "calcified," or "not a nodule"; disagreements were arbitrated by a third radiologist. Ordinal logistic regression was performed as the main statistical test.

**Results**. A total of 592 opacities were included in the observer study. A total of 163/592 classifications (27.5%) required arbitration. A total of 17/592 opacities (2.9%) were excluded from the analysis because they were not considered nodular, were calcified, or all three radiologists disagreed. A total of 366/575 accepted nodules (63.7%) were considered PFNs. A maximum of 10 PFNs were found in one image; 154/324 (47.5%) contained no PFNs. The number of PFNs per subject did not differ between COPD severity groups ($p = 0.50$). PFN short-axis diameter could significantly distinguish between the Mild and Moderate/severe groups, but not between the Healthy and Mild groups ($p = 0.021$).

**Conclusions**. There is no relationship between PFN count and COPD severity. There may be a weak trend of larger intrapulmonary lymph nodes among patients with more advanced stages of COPD.

## INTRODUCTION

Chronic obstructive pulmonary disease (COPD) is an increasingly prevalent disease currently causing about three million deaths per year in 2015, about 5% of all deaths world-wide (*Mathers & Loncar, 2006*; *World Health Organization, 2017*). The clinical diagnosis of COPD is confirmed using spirometry, defined as a forced expiratory volume in 1 s (FEV1) to forced vital capacity (FVC) ratio of less than 0.7 (*Vestbo, 2014*). COPD severity can be further divided into Global Initiative for Chronic Obstructive Lung Disease (GOLD) stages 1 to 4 based on the percentage of the predicted FEV1. More recently, an unclassified GOLD stage better known as ''preserved ratio impaired spirometry'' (PRISm) is also taken into consideration for cases where FEV1/FVC is above 0.7 but FEV1 is less than 80% predicted (*Wan et al., 2011*).

The pathophysiology of COPD is complex and heterogeneous (*Noujeim & Bou-Khalil, 2013*); one of the main mechanisms is immune system-mediated inflammation, which may be associated to the enlargement of mediastinal lymph nodes and development of pulmonary lymphoid follicles (*Habermann & Steensma, 2000*; *Brusselle et al., 2009*; *Kirchner et al., 2010*; *Nin et al., 2016*). However, the role of pulmonary lymphatics in COPD remains poorly understood (*Stump et al., 2017*); the prevalence and size of intrapulmonary lymph nodes had not yet been studied in this patient group. Typical morphological features of these lymph nodes on CT are a diameter of less than 12 mm, an angular, flat, round, or ovoid shape, solid consistency, sharp margins, extending linear opacities, and a location below the level of the carina within 20 mm of the pleura (*Kradin, Spirn & Mark, 1985*; *Bankoff et al., 1996*; *Yokomise et al., 1998*; *Miyake et al., 1999*; *Matsuki et al., 2001*; *Oshiro et al., 2002*; *Hyodo et al., 2004*; *Wang et al., 2013*).

In an attempt to accurately identify benign intrapulmonary lymph nodes from malignant pulmonary nodules on CT, some of these typical lymph node features were used to define the perifissural nodule (PFN), a purely radiologic classification (*Ahn et al., 2010*; *De Hoop et al., 2012*; *Mets et al., 2018*; *Schreuder et al., 2018*). On the premise that more pulmonary inflammation leads to more identifiable intrapulmonary lymph nodes, we hypothesized that more and larger PFNs would be detected as the COPD severity increases.

## METHODS AND MATERIALS

### CT scans and data

CT scans, clinical data, and spirometry outcomes were obtained with approval from the COPDGene study (which was approved by all participating institutional review boards) for the purpose of performing ancillary study number 271 (*Regan et al., 2010*). All participants signed the consent form. A Data Access Investigator Certification Form was signed before the data was provided; ethical board approval for our ancillary study was waived. The

CT protocols can be found in Appendix 1 of the study design (*Regan et al., 2010*). The only subject inclusion criterium was that a baseline and a five-year follow-up CT images were available. We initially calculated a recommended sample size of 39 per group based on a power analysis to detect an effect size of 0.25—moderate according to *Fritz, Morris & Richler (2012)*—with one-way ANOVA of five groups (GOLD stages 0 to 4), where $\alpha = 0.05$ and power $= 0.80$. A pilot study had been performed with a sample size of 200 where about 10% of the scans were missing (results not published); we determined that additional subjects should be included to compensate for this. For our study, 50 subjects were randomly selected from each of the following seven groups (based on classifications at baseline): smokers classified as GOLD stages 0 to 4, smokers classified as PRISm, and never-smoking controls (GOLD 0).

Since 2017, the GOLD strategy documents recommended an "ABCD assessment" where COPD patients are further divided into one of four groups (i.e., A, B, C, or D, from low to high risk) depending on symptoms and exacerbation frequency and severity (*Singh et al., 2019*). Each individuals' category was determined as a parameter for COPD symptom severity. Additionally, two quantitative CT measures of emphysema and chronic bronchitis were obtained using CIRRUS Lung Quantification (Diagnostic Image Analysis Group; Fraunhofer MEVIS): emphysema score and bronchial wall thickness (Pi10) (*Gallardo-Estrella et al., 2017*; *Kim et al., 2014*). Emphysema score was calculated by calculating the percentage of lung voxels below $-950$ HU after resampling the CT images to 3 mm slice thickness, normalization, and bullae analysis (*Gevenois et al., 1996*; *Gallardo-Estrella et al., 2016*). Pi10 is the square root of the airway wall area for a theoretical 10 mm lumen perimeter airway derived using the linear regression of the square root of segmented wall area against the lumen perimeter (*Charbonnier et al., 2019*).

## Pulmonary nodules

The baseline and follow-up CT scans were screened for nodules by a trained medical researcher with support from computer-aided detection software CIRRUS Lung Screening (Diagnostic Image Analysis Group; Fraunhofer MEVIS). Only nodules of longest diameter of at least 3 mm were considered on the premise that the morphological features of smaller nodules would not be perceptible. Size was extracted automatically from the nodule segmentation in the form of volume, longest diameter, and short-axis diameter. Size changes in nodules which were detected in both scans were recorded; if the difference was larger than the known measurement variance of $\pm22.3\%$, the nodule was labelled as growing or shrinking (*De Hoop et al., 2009*).

In the baseline scans, all opacities deemed to be non-calcified nodules were subsequently assessed independently by two radiologists: a general radiologist with over 30 years' experience and a focus on chest radiology including 5 years of lung cancer screening experience, and a chest radiologist with more than 20 years of experience in thoracic imaging. The classification options were as follows: "PFN", "non-PFN solid", "part-solid", "non-solid", "calcified", and "not a nodule". Disagreements were arbitrated by a third radiologist with over 30 years' experience.
All radiologists were familiar with the morphology of PFNs as described in the literature (*Ahn et al., 2010*; *De Hoop et al., 2012*; *Mets et al., 2018*; *Schreuder et al., 2018*). Non-calcified solid nodules with sharp margins and an oval or polygonal shape were eligible as PFNs. Despite the nomenclature, PFNs do not need to be directly adjacent to fissures (*De Hoop et al., 2012*; *Mets et al., 2018*; *Schreuder et al., 2018*). Readers were instructed to only classify a nodule as a PFN if highly certain that it was a lymph node. All opacities agreed on as "calcified" or "not a nodule" and opacities which were labelled differently by all readers were excluded from the statistical analysis.

## Statistical analysis

The observer study outcomes were imported into Microsoft Excel$^{TM}$ 365 and subsequently to R statistical analysis package version 3.4.3. The number of PFNs, non-PFN solid nodules, part-solid nodules, and non-solid nodules were counted per scan. Nodule characteristics were compared between PFNs and non-PFN solid nodules using the Mann–Whitney $U$ test for continuous variables and the chi-squared test for categorical variables. Inter-reader agreement was measured using Cohen's kappa with all six nodule classification options and with two classes (PFN or not) (*Cohen, 1960*).

Regression analysis was performed to test whether the number and size (volume, longest diameter, and short-axis diameter) of PFNs detected could predict COPD severity. Ordered logistic regression (R function "polr" from package "MASS") was used to model ordinal dependent variables (GOLD staging and ABCD assessment), and linear regression for continuous variables (emphysema score and Pi10). This analysis was performed using four outcome measures of COPD: GOLD severity groups ("Healthy" [GOLD 0 never and ever smokers], "Mild" [GOLD 1 and 2 and PRISm], and "Moderate/severe" [GOLD 3 and 4]), ABCD assessment, emphysema score ($\ln(x+1)$ transformed due to non-linearity of residuals), and Pi10.

PFN count was analysed as an independent variable at the subject level; PFN sizes were modelled at the nodule level. This analysis was performed with and without adjustment for the baseline variables age, sex (female VS male), race (non-Hispanic Black VS non-Hispanic White), and smoking status (current VS former VS never). Analysis of GOLD severity groups was not adjusted for smoking status because this variable was used to define the groups. Statistical significance was defined as $p < 0.0253$ after applying the Šidák correction for multiple comparisons (calculated for only two hypotheses because the measures of PFN size and measures of COPD severity were considered complementary) (*Šidák, 1967*; *Rothman, 1990*).The Shapiro–Wilk test was used to test for normality; all numeric outcomes were not normally distributed ($p < 0.001$) which justified performing the analyses using non-parametric tests.

Post hoc analysis was performed using univariable linear regression to associate more variables of interest—age, sex, race (White and non-Hispanic Black), educational level, smoking status, smoking duration, exacerbation frequency, at least one severe exacerbation, Modified Medical Research Council (mMRC) Dyspnea Scale, exposure to a dusty job (for at least one year), and exposure to fumes at work (for at least one year)—to PFN count and PFN volume. mMRC score and educational level was transformed into a numeric trend;

for the latter, 1 = "8th grade or less", 2 = "high school, no diploma", 3 = "high school graduate or General Educational Development certificate", 4 = "some college or technical school, no degree", 5 = "college or technical school graduate (Bachlor's or Associate degree)", and 6 = Master's or Doctoral degree". For these 22 comparisons, the level of significance was lowered to 0.0023 (*Šidák, 1967*).

## RESULTS

### Observer study

The inclusion criteria and study design of COPDGene are described elsewhere (*Regan et al., 2010*). In short, it is a multicenter observational study collecting clinical, functional, imaging, and genetic data related to COPD. Ten thousand non-Hispanic Whites and Blacks between 45 and 80 years with and without COPD were enrolled at baseline (between January 2008 and June 2011). All participants required at least 10 pack-years of smoking, except for the control group of 133 never-smokers. A second screening round was performed five years after baseline.

The subject inclusion process is shown in Fig. 1. Of the 350 subjects included, 26 (7.4%) were excluded because the baseline and/or follow-up scans were missing from the database. The baseline characteristics across GOLD stages are shown in Table 1. Out of 324 baseline scans, 592 nodules were included in the observer study; 211 scans (65.1%) contained at least one nodule. After the observer study had been performed independently by Readers A and B, there were 163 disagreements (27.5%) which were arbitrated by Reader C. After arbitration, six opacities (6/592, 1.0%) were not considered nodules, five (5/592, 0.8%) were calcified, and six (6/592, 1.0%) were given different classifications by all three readers; these were excluded from the statistical analysis. Fourteen subjects (14/324, 4%) had missing values for mMRC score (therefore also lacking an ABCD assessment), 10 (3%) had missing values for exposure to a dusty job (for at least one year), and 20 (6%) for exposure to fumes at work (for at least one year); these were excluded from the relevant analyses.

### Inter-rater agreement

Agreement in the form of Cohen's kappa between readers A and B for 592 nodules was 0.50 (95% confidence interval: 0.43 to 0.56) with six classification options and 0.54 (0.47 to 0.61) when only distinguishing between a PFN and all other classification options, interpretable as moderate agreement. Among the 163 disagreements, the six-class kappa between Readers A and C and Readers B and C were 0.06 ($-$0.06 to 0.17) and 0.17 (0.05 to 0.29), respectively; the two-class kappa were 0.11 ($-$0.04 to 0.26) and 0.10 ($-$0.05 to 0.25), respectively; these values indicate that there was no significant agreement (*Cohen, 1960*).

### Nodule characteristics

The characteristics of the nodules analysed are summarized in Table 2. Most nodules were classified as PFNs (366/575, 63.7%); Fig. 2 is an example of what is considered a typical PFN. Compared to non-PFN solid nodules (180/575, 31.3%), PFNs were more often

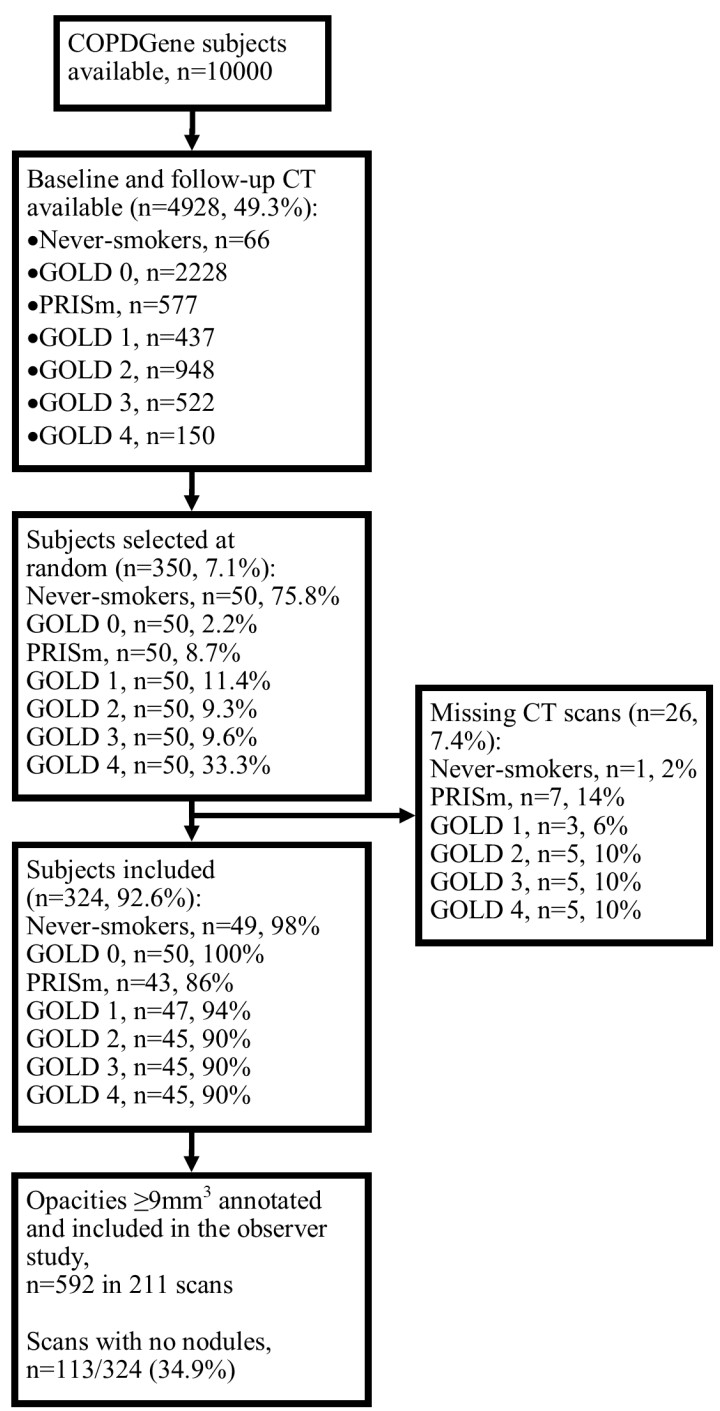

**Figure 1 Subject selection process.** Abbreviations: PRISm, preserved ratio impaired spirometry.

smaller, flatter in shape, found in the middle or lower lobes, and remained the same size at follow-up; note that no nodule in the study was found to have a volume doubling time of less than 600 days. A much smaller number of subsolid nodules (29/575, 5.0%) was detected.

### Perifissural nodule count

The prevalence and distribution of PFNs across COPD GOLD groups are shown in Table 3 and Fig. 3, and the data is provided as a supplementary file which also includes basic characteristics. Roughly one-third of the scans contained no nodules (118/324, 36.4%) and half contained no PFNs (154/324, 47.5%). The most PFNs found in one scan was 10. The average number of PFNs ranged from 0.92 (1.50) in the GOLD 0 ever-smokers group to 1.40 (1.95) in the GOLD 4 group. PFN count was not a statistically significant predictor for GOLD severity groups ($p = 0.50$), ABCD assessment ($p = 0.94$), emphysema score ($p = 0.13$), or Pi10 ($p = 0.70$). The respective $p$ values were 0.33, 0.69, 0.52, and 0.94 after adjusting for other factors. There were no significant associations of PFN count with other variables (Table 4), though there may be a trend where non-Hispanic Blacks have fewer PFNs than Whites ($p = 0.005$).

### Perifissural nodule size

The distribution of PFN sizes across GOLD stages are listed in Table 3; Fig. 4 shows a bar plot of the PFN density (number of PFNs in the size category divided by the number of PFNs in the GOLD stage) versus volume per GOLD stage. The median volume, longest diameter, and short-axis diameter of the PFNs analysed was 22.3 mm$^3$ (interquartile range = 18 mm$^3$), 4.8 mm (1.7 mm), and 3.2 mm (1.0 mm), respectively.

Volume, longest diameter, and short-axis diameter were larger on average in subjects more severe GOLD stages ($p = 0.050$, $p = 0.023$, and $p = 0.021$, respectively). The differences were largest between the Mild and Moderate/severe groups; the difference between the Healthy and Mild groups were not significant for longest diameter ($p = 0.51$) and short-axis diameter ($p = 0.83$). Univariable analysis for ABCD assessment showed a stronger trend, where larger PFNs were usually found in symptomatic participants at risk of exacerbation (volume $p = 0.002$; longest diameter $p = 0.011$; short-axis $p = 0.012$). Only PFN volume was a significant predictor of emphysema score ($p = 0.043$; longest diameter $p = 0.053$; volume $p = 0.07$, respectively). None of the PFN sizes could be used as predictors of Pi10 ($p = 0.14$, $p = 0.19$, and $p = 0.23$, respectively).

After adjusting for other factors (excluding smoking status), only short-axis diameter remained a statistically significant predictor of GOLD severity group ($p = 0.012$; volume $p = 0.050$; longest diameter $p = 0.07$). The respective $p$ values for adjusted analysis of ABCD assessment are 0.076, 0.017, and 0.053. A stronger correlation was found (similar to the univariable analyses) when excluding smoking status from the list of adjustment factors (volume $p = 0.002$; longest diameter $p = 0.011$; short-axis $p = 0.011$). The most statistically significant post hoc finding was the positive trend between PFN volume and mMRC score ($p = 0.004$), though this only explained 2% of the variation (adjusted $r$-squared).

## DISCUSSION

The primary aim of this study was to determine whether a larger number or size of PFNs detected in a CT scan was associated to a higher level of COPD severity. Four measures of COPD—one functional, one symptomatic, and two quantitative—were considered. The reasoning was that greater COPD severity is associated with more inflammation,

**Table 1  Baseline subject descriptive statistics.** For emphysema score, the values given are medians followed by the 25th and 75th percentiles in parentheses. The remaining continuous variables are given as means followed by the standard deviations in parentheses. The $p$ values of all continuous variables were calculated using the linear regression $F$ test. For factors, values in parentheses are percentages; the $p$ values were calculated from the chi-square test.

| GOLD stage | 0 (NS), n = 49 | 0 (ES), n = 50 | PRISm, n = 43 | 1, n = 47 | 2, n = 45 | 3, n = 45 | 4, n = 45 | P value | Total, n = 324 |
|---|---|---|---|---|---|---|---|---|---|
| Age (years) | 64.3 (9.6) | 63.9 (8.7) | 60.9 (7.5) | 67.9 (8.2) | 68.3 (7.5) | 68.5 (8.5) | 66.9 (8.0) | <0.001 | 65.8 (8.7) |
| Sex (female) | 32 (9.9) | 30 (9.3) | 19 (5.9) | 24 (7.4) | 20 (6.2) | 16 (4.9) | 16 (4.9) | 0.021 | 157 (48.5) |
| Race (non-Hispanic Black) | 4 (1.2) | 9 (2.8) | 20 (6.2) | 8 (2.5) | 9 (2.8) | 10 (3.1) | 7 (2.2) | 0.001 | 67 (20.7) |
| Smoking status | | | | | | | | <0.001 | |
| Current | 0 (0) | 19 (5.9) | 21 (6.5) | 15 (4.6) | 17 (5.2) | 9 (2.8) | 3 (0.9) | | 84 (25.9) |
| Former | 0 (0) | 31 (9.6) | 22 (6.8) | 32 (9.9) | 28 (8.6) | 36 (11.1) | 42 (13) | | 191 (59) |
| Never | 49 (15.1) | 0 (0) | 0 (0) | 0 (0) | 0 (0) | 0 (0) | 0 (0) | | 49 (15.1) |
| Emphysema score | 0.1 (0.0–0.1) | 0.1 (0.0–0.2) | 0.0 (0.00–0.1) | 0.6 (0.1–3.5) | 0.5 (0.1–2.1) | 3.9 (0.2–14.4) | 15.2 (4.2–31.2) | <0.001 | 4.6 (9.5) |
| Pi10 | 1.73 (0.29) | 2.01 (0.49) | 2.57 (0.53) | 2.07 (0.51) | 2.67 (0.52) | 2.96 (0.62) | 2.88 (0.45) | <0.001 | 2.39 (0.66) |
| ABCD assessment | | | | | | | | <0.001 | |
| A | 48 (15.5) | 38 (12.3) | 21 (6.8) | 34 (11.0) | 21 (6.8) | 12 (3.9) | 1 (0.3) | | 175 (54.0) |
| B | 0 (0) | 9 (2.9) | 14 (4.5) | 8 (2.6) | 15 (4.8) | 16 (5.2) | 19 (6.1) | | 81 (25.0) |
| C | 0 (0) | 2 (0.6) | 4 (1.3) | 3 (1.0) | 1 (0.3) | 1 (0.3) | 0 (0) | | 11 (3.4) |
| D | 0 (0) | 1 (0.3) | 4 (1.3) | 2 (0.6) | 8 (2.6) | 16 (5.2) | 12 (3.9) | | 43 (13.3) |

**Notes.**

Abbreviations: ES, ever-smoker; NS, never-smoker; Pi10, bronchial wall thickness; PRISm, preserved ratio impaired spirometry.

**Table 2  Nodule characteristics.** For continuous variables, the values given are medians followed by the 25th and 75th percentiles in parentheses and the *p* values were calculated from the Mann-Whitney *U* test. For factors, values in parentheses are percentages and the *p* values were calculated from the chi-squared test. The statistical tests were only performed comparing the "PFN" to the "non-PFN solid" groups.

| Characteristic | PFN, n = 366 | Non-PFN solid, n = 180 | P value | Part-solid, n = 2 | Non-solid, n = 27 |
|---|---|---|---|---|---|
| Volume (mm³) | 22 (17–35) | 32 (18–80) | <0.001 | 3594 (1836–5351) | 92 (55–220) |
| Longest diameter (mm) | 4.8 (4.0–5.7) | 4.9 (4.0–6.8) | 0.17 | 19.1 (12.8–25.5) | 6.3 (5.6–10.3) |
| Short-axis diameter (mm) | 3.2 (2.7–3.7) | 3.6 (3.0–4.7) | <0.001 | 10.3 (7.8–12.8) | 4.6 (3.7–6.8) |
| Lobe location | | | 0.001 | | |
| *Right upper* | 63 (17.2) | 35 (19.4) | | 1 (50.0) | 12 (44.4) |
| *Right middle* | 57 (15.6) | 8 (4.4) | | 0 (0.0) | 1 (3.7) |
| *Right lower* | 91 (24.9) | 45 (25.0) | | 0 (0.0) | 6 (22.2) |
| *Left upper* | 66 (18.0) | 50 (27.8) | | 0 (0.0) | 6 (22.2) |
| *Left lower* | 89 (24.3) | 42 (23.3) | | 1(50.0) | 2 (7.4) |
| Five-year change | | | <0.001 | | |
| *Regression* | 115 (31.4) | 98 (54.4) | | 0 (0) | 7 (25.9) |
| *Stable* | 175 (47.8) | 56 (31.1) | | 1 (50.0) | 8 (29.6) |
| *Growth* | 76 (20.8) | 26 (14.4) | | 1 (50.0) | 12 (44.4) |

**Notes.**
Abbreviations: PFN, perifissural nodule.

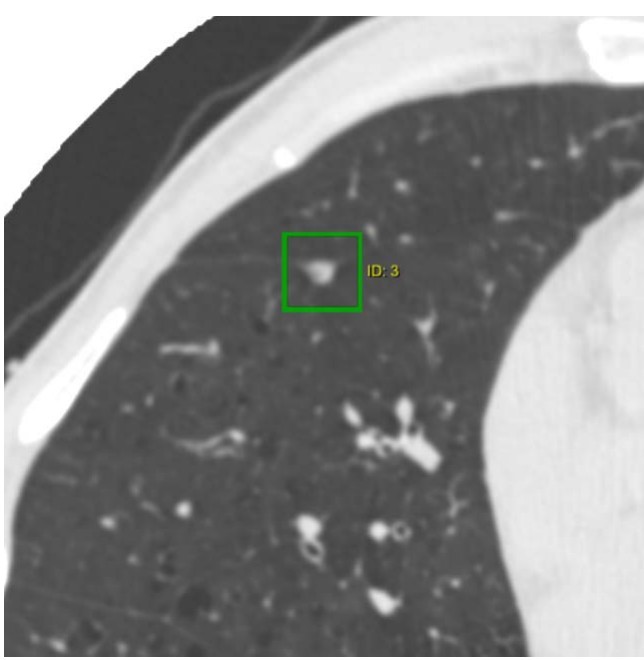

**Figure 2  CT image of a typical perifissural nodule attached to a fissure.** The nodule is a well-defined non-calcified solid nodule which is triangular in shape with a flattened side adjacent to a pulmonary fissure.

which is associated with an overactive immune system characterized by more enlarged intrapulmonary lymph nodes identifiable on CT as PFNs. We found that PFN count did not differ across subjects with different levels of COPD severity. However, there was a

**Table 3 Prevalence of nodules and perifissural nodules across COPD GOLD stages.** The baseline GOLD groups were combined to form the "Healthy" (GOLD 0 never- and ever-smokers), "Mild" (PRISm and GOLD 1 and 2), and "Moderate/severe" (GOLD 3 and 4) groups. The number of scans and PFNs are given in frequencies and percentages. The number of PFNs per scan is given in means and standard deviations. The measures of size are given in medians and 25 th and 75 th percentiles. *P* values were obtained by performing ordered logistic regression between the Healthy, Mild, and Moderate/severe groups to analyze whether the number or size of PFNs could predict COPD severity.

| COPD severity | No. baseline scans | No. PFNs | No. PFNs per scan | PFN volume (mm³) | PFN longest diameter (mm) | PFN short-axis diameter (mm) |
|---|---|---|---|---|---|---|
| *Healthy* | *99 (31)* | *102 (28)* | *1.03 (1.37)* | *21 (16–33)* | *4.6 (3.9–5.3)* | *3.2 (2.6–3.6)* |
| GOLD 0 (NS) | 49 (15) | 56 (15) | 1.14 (1.24) | 20 (16–29) | 4.5 (3.9–5.2) | 3.0 (2.6–3.4) |
| GOLD 0 (ES) | 50 (15) | 46 (13) | 0.92 (1.50) | 22.2 (17.0–39.8) | 4.0 (4.8–5.4) | 3.2 (2.4–3.8) |
| *Mild* | *135 (42)* | *156 (43)* | *1.16 (1.89)* | *22 (17–32)* | *5.0 (4.1–5.8)* | *3.1 (2.7–3.8)* |
| PRISm | 43 (13) | 40 (11) | 0.93 (1.39) | 18 (17–27) | 5.0 (4.3–5.9) | 3.1 (2.7–4.0) |
| GOLD 1 | 47 (15) | 63 (17) | 1.34 (2.08) | 21 (16–31) | 4.7 (3.9–5.7) | 2.9 (2.7–3.4) |
| GOLD 2 | 45 (14) | 53 (14) | 1.18 (2.10) | 25 (18–38) | 5.1 (4.1–5.8) | 3.4 (2.9–3.9) |
| *Moderate/severe* | *90 (28)* | *108* | *1.20 (1.70)* | *26 (18–47)* | *5.0 (4.2–5.9)* | *3.3 (2.8–3.9)* |
| GOLD 3 | 45 (14) | 45 (12) | 1.00 (1.40) | 27 (18–48) | 5.0 (4.2–5.5) | 3.4 (3.0–3.9) |
| GOLD 4 | 45 (14) | 63 (17) | 1.40 (1.95) | 26 (17–46) | 5.0 (3.9–6.0) | 3.2 (2.7–3.8) |
| **Total** | **324 (100)** | **366 (100)** | **1.13 (1.69)** | **22 (17–35)** | **4.8 (4.0–5.7)** | **3.2 (2.7–3.7)** |
| *P* value | – | – | 0.50 | 0.07 | 0.07 | 0.07 |

**Notes.**

Abbreviations: no., number of; ES, ever-smoker; NS, never-smoker; PFN, perifissural nodule; PRISm, preserved ratio impaired spirometry.

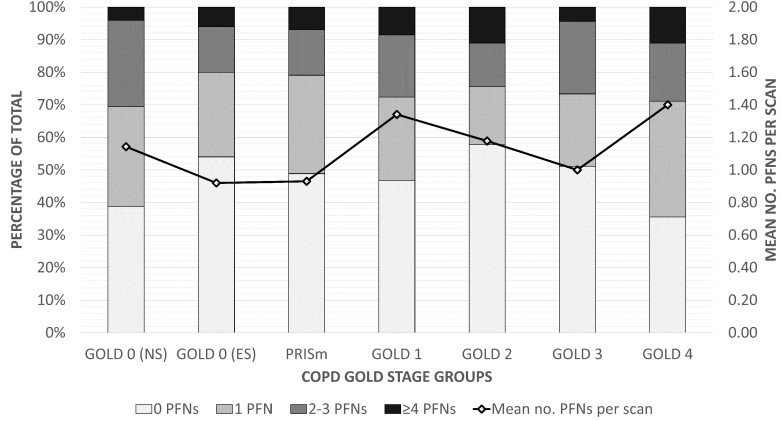

**Figure 3 Distribution of PFNs across COPD GOLD stages.** A 100% stacked bar chart illustrating the number of PFNs distributed across scans within each GOLD group. The line represents the average number of PFNs per scan for each group (given on the right axis). Abbreviations: ES, ever-smoker; no., number of; PFN, perifissural nodule; PRISm, preserved ratio impaired spirometry.

positive trend in PFN size for GOLD severity and ABCD assessment groups, also when adjusted for age, sex, and race. The differences in PFN sizes were less evident between the Healthy and Mild GOLD groups and the B and C ABCD groups. Overall, the small group differences and the high variance prevent PFN features from being clinically relevant measures: no PFNs were detected in almost half (154/324, 47.5%) of all scans while the other half contained a range of one to 10 PFNs (Table 3).

The ABCD groups are determined based on the mMRC Dyspnea Scale and exacerbation frequency and severity. In response to the positive findings related to ABCD assessment and

**Table 4   Perifissural nodule count and volume association with other variables.** Univariable linear regression between the number and volume of PFNs and other variables of interest. The $\beta$ coefficient represents the magnitude of the association (per unit for continuous variables). mMRC score and educational level was transformed into a numeric variable (for the latter, 1, 8th grade or less; 2, high school, no diploma; 3, high school graduate or General Educational Development certificate; 4, some college or technical school, no degree; 5, college or technical school graduate; and 6, Master's or Doctoral degree.). Emphysema score was $\ln(x+1)$ transformed. A $p$ value of less than 0.0023 was considered significant.

| Variable | No. PFNs per scan | | PFN volume (mm³) | |
|---|---|---|---|---|
| | $\beta$ coefficient | P value | $\beta$ coefficient | P value |
| Age (years) | 0.02005 | 0.06 | −0.2286 | 0.19 |
| Sex (female) | 0.1563 | 0.41 | −1.871 | 0.55 |
| Race (non-Hispanic Black) | −0.6527 | 0.005 | 6.889 | 0.16 |
| Educational level (per unit) | −0.03459 | 0.65 | −1.347 | 0.29 |
| Smoking status | | 0.21 | | 0.11 |
| *Current* | Ref. | Ref. | Ref. | Ref. |
| *Former* | 0.3889 | 0.08 | −1.459 | 0.71 |
| *Never* | 0.2857 | 0.35 | −10.042 | 0.058 |
| Smoking duration (years) | 0.004093 | 0.40 | 0.03057 | 0.69 |
| Exacerbation frequency (count) | 0.05789 | 0.54 | 2.129 | 0.12 |
| At least one severe exacerbation | 0.3229 | 0.28 | 1.125 | 0.80 |
| mMRC score (per unit) | −0.03202 | 0.65 | 3.064 | 0.004 |
| Exposure to dusty job | −0.1439 | 0.46 | 6.485 | 0.043 |
| Exposure to fumes at work | −0.02020 | 0.92 | 4.170 | 0.18 |
| Emphysema score (per unit) | 0.1237 | 0.13 | 2.631 | 0.043 |
| Pi10 (per unit) | −0.05412 | 0.70 | 0.2291 | 0.14 |
| ABCD assessment (ordered A–D) | −0.004985 | 0.94 | 0.1017 | 0.002 |

**Notes.**

Abbreviations: mMRC, Modified Medical Research Council; no., number of; PFN, perifissural nodule; Ref., reference variable.

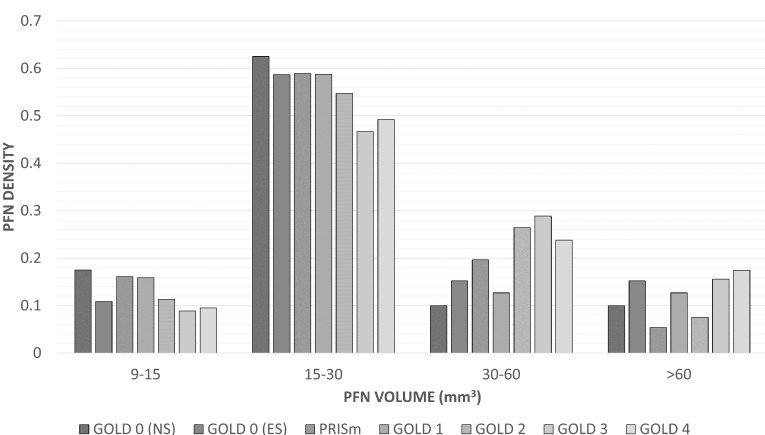

**Figure 4   Density of perifissural nodules across short-axis diameters and GOLD stages.** A bar chart of the density (number of PFNs in the size category divided by the number of PFNs in the GOLD stage) of perifissural nodules grouped by volume and GOLD stage. The association between GOLD severity groups (Healthy, GOLD 0 never- and ever-smokers; Mild, PRISm and GOLD 1 and 2; Moderate/severe, GOLD 3 and 4) was not considered statistically significant with ($p = 0.050$) and without adjusting for other factors ($p = 0.050$). Abbreviations: PFN, perifissural nodule; PRISm, preserved ratio impaired spirometry.

PFN size, these three variables were assessed to determine whether similar effects could be seen in the individual components. Table 4 shows that dyspnea symptoms have a stronger association with larger PFN volume ($p = 0.004$) than the measures of exacerbation risk ($p \geq 0.12$). Other post hoc analysis showed that there was a negative trend between the PFN count and non-Hispanic Black race ($p = 0.005$), and that PFN volume was greater in current smokers compared to never-smokers ($p = 0.058$) and in those which had been exposed to a dusty work environment ($p = 0.043$) (Table 4). However, considering the large number of comparisons, these may be false positive findings (for which the threshold for significance was set at 0.0023). Also, an association of having 0.6527 fewer PFNs per scan is not clinically useful. No other trends were found between PFN count or volume and other variables of interest.

This is not the first study to investigate a relationship between the lymphatic system and COPD. Brusselle and colleagues found a higher prevalence of peribronchial and parenchymal lymphoid follicles in severe COPD cases (*Brusselle et al., 2009*), and Kirchner and colleagues found that about half of COPD patients had enlarged mediastinal lymph nodes (*Kirchner et al., 2010*); they found no trend across increasing GOLD stages. Notably, the literature defines enlarged mediastinal lymph nodes as those 10 mm or larger in short-axis diameter (*Kirchner et al., 2010*; *Nin et al., 2016*). Though the same threshold would not be useful for intrapulmonary lymph nodes—only 1% of PFNs (4/366) fit this criterium—we also found short-axis diameter to be better at distinguishing GOLD severity groups than volume and longest diameter. One possible explanation would it is a better indicator of how "round" a PFN is: a long but flat lymph node may be less active than a more swollen PFN. PFN volume may also reflect this to a lesser extent.

Our findings agree with those of Mets and colleagues who investigated PFN malignancy chance in a clinical setting (*Mets et al., 2018*). They found no significant differences in visually assessed presence or extent of emphysema (4-point ordinal scale), presence of bronchial wall thickening, vascular calcifications (4-point scale), or presence of hilar/mediastinal lymphadenopathy between patients with and without PFNs. It was expected that bronchial wall thickness would have a greater association to PFN count and size because it is a measure of bronchitis; it is unclear why a stronger association was found between PFN size and GOLD stages—a measure of ventilation.

The PFN characteristics recorded (Table 2) are also comparable to those of previous studies (*Ahn et al., 2010*; *De Hoop et al., 2012*; *Mets et al., 2018*; *Schreuder et al., 2018*). Our results support the notion that the average PFN is significantly different from the average non-PFN solid nodule. One of the few consistent criteria in defining a nodule as a PFN is that it must be of solid consistency. Less consistent with previous reports is the relatively high number of PFNs found in this study (366/575, 63.7%). As PFNs tend to be smaller, this may be explained by the inclusion of very small nodules in our study, though features are also more difficult to distinguish among smaller nodules. However, 78 of the 575 nodules (13.6%) were marked with a comment on by at least one reader that it was too small to be certain that it was a PFN; this indicates that the readers did not simply classify nodules as PFNs based on size or benignity likelihood. Although such small nodules are clinically irrelevant when considering malignancy probability, it could have been relevant for the

purpose of this study. That half of the PFNs grew or shrank in size has also been previously reported and does not influence its chance of being a malignancy (*De Hoop et al., 2012*).

Variation in nodule detection rates between radiologists can be very high (*Rothman, 1990*); this study largely avoids this by preselecting nodules as well as relying on agreement among three independent readers. The inter-radiologist agreement between the two readers who classified all nodules in the observer study was 0.54 (95% confidence interval: 0.47 to 0.61) when distinguishing between PFNs and any other classification; this is within the range reported by a previous study on PFN reader variability among six readers (*Schreuder et al., 2018*). This indicates that some readers would systematically detect more PFNs than other readers. On the other hand, our consensus reader had a similar level of slight agreement (not statistically significant) with both initial readers, rejecting the possibility of preferential treatment (despite blinding). No official PFN definition was provided, which may have played a role in the observer variability and restricts reproducibility for future studies.

There are several other limitations to our study. There is selection bias in that all subjects had a baseline and follow-up scan, meaning that the subjects must have survived within the period of five years in between. This was to ensure that nodule change could be recorded; it is unknown whether the results would have differed without this selection criterium.

No pathological confirmation was available to confirm which opacities were lymph nodes. Due to the survival time and no nodule having a volume doubling time of less than 600 days, it can be said with high certainty that cancers did not reside among them. Besides lymph nodes, many solid nodules are expected to be granulomas which also originate in response to inflammation but do not typically have the shape of lymph nodes on CT (*Hyodo et al., 2004*; *Honma, Nelson & Murray, 2007*). To account for the possibility that the true number of intrapulmonary lymph nodes was greater than the number of PFNs counted, the same analysis was performed on all solid nodules, but no significant trends were found (results not included).

In conclusion, there was no association found between the number of PFNs detected in a chest CT scan and the level of COPD severity. However, there may be a weak trend of larger intrapulmonary lymph nodes in moderate to severe stages of COPD based on functional, symptomatic, and quantitative CT measures. Nevertheless, it would likely not be a clinically useful biomarker: Variance was large across subjects, where about half of the scans did not contain any PFNs while as many as 10 PFNs were detected in one CT image. Additionally, classifying nodules as PFNs without clear criteria was again shown to be a task with much disagreement among experienced radiologists.

**List of abbreviations**

| | |
|---|---|
| **COPD** | chronic obstructive pulmonary disease |
| **GOLD** | Global Initiative for Chronic Obstructive Lung Disease |
| **FEV1** | orced expiratory volume in 1 second |
| **FVC** | forced vital capacity |
| **mMRC** | Modified Medical Research Council |
| **PFN** | perifissural nodule |
| **Pi10** | bronchial wall thickness |
| **PRISm** | preserved ratio impaired spirometry |

## ACKNOWLEDGEMENTS

The content is solely the responsibility of the authors and does not necessarily represent the official views of the National Heart, Lung, and Blood Institute or the National Institutes of Health. The members of the COPDGene Investigators are: Administrative Center: James D. Crapo, MD (PI); Edwin K. Silverman, MD, PhD (PI); Barry J. Make, MD; Elizabeth A. Regan, MD, PhD. Genetic Analysis Center: Terri Beaty, PhD; Ferdouse Begum, PhD; Peter J. Castaldi, MD, MSc; Michael Cho, MD; Dawn L. DeMeo, MD, MPH; Adel R. Boueiz, MD; Marilyn G. Foreman, MD, MS; Eitan Halper-Stromberg; Lystra P. Hayden, MD, MMSc; Craig P. Hersh, MD, MPH; Jacqueline Hetmanski, MS, MPH; Brian D. Hobbs, MD; John E. Hokanson, MPH, PhD; Nan Laird, PhD; Christoph Lange, PhD; Sharon M. Lutz, PhD; Merry-Lynn McDonald, PhD; Margaret M. Parker, PhD; Dmitry Prokopenko, PhD; Dandi Qiao, PhD; Elizabeth A. Regan, MD, PhD; Phuwanat Sakornsakolpat, MD; Edwin K. Silverman, MD, PhD; Emily S. Wan, MD; Sungho Won, PhD; Imaging Center: Juan Pablo Centeno; Jean-Paul Charbonnier, PhD; Harvey O. Coxson, PhD; Craig J. Galban, PhD; MeiLan K. Han, MD, MS; Eric A. Hoffman, Stephen Humphries, PhD; Francine L. Jacobson, MD, MPH; Philip F. Judy, PhD; Ella A. Kazerooni, MD; Alex Kluiber; David A. Lynch, MB; Pietro Nardelli, PhD; John D. Newell, Jr., MD; Aleena Notary; Andrea Oh, MD; Elizabeth A. Regan, MD, PhD; James C. Ross, PhD; Raul San Jose Estepar, PhD; Joyce Schroeder, MD; Jered Sieren; Berend C. Stoel, PhD; Juerg Tschirren, PhD; Edwin Van Beek, MD, PhD; Bram van Ginneken, PhD; Eva van Rikxoort, PhD; Gonzalo Vegas Sanchez-Ferrero, PhD; Lucas Veitel; George R. Washko, MD; Carla G. Wilson, MS; PFT QA Center, Salt Lake City, UT: Robert Jensen, PhD; Data Coordinating Center and Biostatistics, National Jewish Health, Denver, CO: Douglas Everett, PhD; Jim Crooks, PhD; Katherine Pratte, PhD; Matt Strand, PhD; Carla G. Wilson, MS; Epidemiology Core, University of Colorado Anschutz Medical Campus, Aurora, CO: John E. Hokanson, MPH, PhD; Gregory Kinney, MPH, PhD; Sharon M. Lutz, PhD; Kendra A. Young, PhD; Mortality Adjudication Core: Surya P. Bhatt, MD; Jessica Bon, MD; Alejandro A. Diaz, MD, MPH; MeiLan K. Han, MD, MS; Barry Make, MD; Susan Murray, ScD; Elizabeth Regan, MD; Xavier Soler, MD; Carla G. Wilson, MS; Biomarker Core: Russell P. Bowler, MD, PhD; Katerina Kechris, PhD; Farnoush Banaei-Kashani, PhD.

### Funding
The COPDGene® project was supported by Award Number U01 HL089897 and Award Number U01 HL089856 from the National Heart, Lung, and Blood Institute. COPDGene® is also supported by the COPD Foundation through contributions made to an Industry Advisory Board comprised of AstraZeneca, Boehringer Ingelheim, GlaxoSmithKline, Novartis, Pfizer, Siemens and Sunovion. The funders had no role in study design, data collection and analysis, decision to publish, or preparation of the manuscript.

### Grant Disclosures
The following grant information was disclosed by the authors:
The COPDGene® project: U01 HL089897.
National Heart, Lung, and Blood Institute: U01 HL089856.
COPD Foundation.

### Competing Interests
Colin Jacobs received a research grant from MeVis Medical Solutions AG; Mathias Prokop received research funding from Toshiba, the speaker's bureau at Bracco, Bayer, Toshiba, and Siemens, and performed a department spin-off at Thiroux (no personal financial interest); Bram van Ginneken received royalties from MeVis Medical Solutions and Delft Imaging Systems, and is a co-founder and shareholder of Thirona. David A. Lynch received Grant support from the NIH/NHLBI, and income as a consultant for Boehringer Ingelheim and Parexel. Anton Schreuder, Ernst T. Scholten, and Cornelia M. Schaefer-Prokop do not have any competing interests.

### Author Contributions
- Anton Schreuder conceived and designed the experiments, performed the experiments, analyzed the data, prepared figures and/or tables, authored or reviewed drafts of the paper, and approved the final draft.
- Colin Jacobs and Ernst T. Scholten performed the experiments, authored or reviewed drafts of the paper, and approved the final draft.
- Mathias Prokop and David A. Lynch conceived and designed the experiments, performed the experiments, authored or reviewed drafts of the paper, and approved the final draft.
- Bram van Ginneken conceived and designed the experiments, authored or reviewed drafts of the paper, and approved the final draft.
- Cornelia M. Schaefer-Prokop performed the experiments, authored or reviewed drafts of the paper, and approved the final draft.

### Human Ethics
The following information was supplied relating to ethical approvals (i.e., approving body and any reference numbers):

Approval to use the data was obtained from the COPDGene study; ethical board approval was not required as the data were anonymized.

## Data Availability

The patient characteristics, GOLD stages, ABCD assessments, quantitative CT measures of COPD, and the number and types of nodules detected (by majority vote) are available as a Supplemental File.

The ID numbers are unique per participant and do not correspond to the ID numbers in the COPDGene data set.

Raw COPDGene data is available by request per NCBI regulations: https://www.ncbi.nlm.nih.gov/projects/gap/cgi-bin/study.cgi?study_id=phs000179.v6.p2.

## Supplemental Information

Supplemental information for this article can be found online at http://dx.doi.org/10.7717/peerj.9166#supplemental-information.

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
