# Peer review of "Association between the number and size of intrapulmonary lymph nodes and chronic obstructive pulmonary disease severity"

_PeerJ, doi:10.7717/peerj.9166_

## Round 0.1 · original submission · Major Revisions

Your manuscript has been reviewed and requires modifications prior to making a decision. The comments of the reviewer(s) are included at the bottom of this letter. Reviewers indicated that the design of the study and the methods section should be improved. Review 1 also has a concern about ethical issues. I agree with the evaluation and I would, therefore, request for the manuscript to be revised accordingly. I would also like to suggest the following change:

Did you check the normality assumption of the data? If yes, which normality test was used? Please provide the name of the statistical test.

·

Basic reporting

professional English was used, only some minor mistakes are present, also compare minor issues

Experimental design

see below

Validity of the findings

see below

Additional comments

In the present manuscript, Schreuder et al report on the impact of perifissural lymphatic nodule (PFN) characteristics on COPD severity. The manuscript is based on a large clinical study (COPDGene) and 50 patients per COPD group were randomly recruited from this big database resulting in 350 analyzed study subjects.
In the end, no association of the number of PFN and COPD severity was found, however, a tendency towards lymph node size (in short diameter) and more advanced disease was suggested with regard to PFNs.
In summary, this study is about a relevant and interesting topic and also the study design involving the above mentioned large clinical trial data is a promising approach. However, some major issues have to be addressed before the present manuscript can be recommended for publication.

1) When studying the (complex) impact of inflammation on COPD severity one has to take into account that not a single marker like e.g. radiological characteristic of only one lymph node subpopulation/station is representative for such a complex disease involving the whole lung and its associated thoracic structures and furthermore the whole immune system.

2) Thus one might be interested in:

A) Systemic inflammatory parameters (blood parameters)
B) The whole lymphatic system including mediastinal as well as hilar lymph nodes
C) Local (in the lung and adjacent tissue) inflammation

3) Since the COPDGene database should offer the possibility not to only study PFNs but also the aforementioned local and systemic inflammatory parameters, a more comprehensive analysis with regard to inflammation would be of interest and worth to be published. One – very special – single marker without a well-documented rational behind it might not be representative for the whole pathogenesis behind a – heterogeneous and complex chronic – disease like COPD.

4) Was really no ethical approval required?

5) Is it valid to summarize PRISm together with GOLD 1 & 2 in the mild group?

6) What about the follow up? Was the dynamic of the lymphadenopathy relevant for the clinical curse? Was enlargement of lymph nodes associated with progression of disease over the 5 year period? You have this 5 year follow up and should use it, would be a clear strength of the study to use the 5 year data to test your marker(s) also as follow up markers!


Minor Issues:
Abstract line 45 should be associated with lymphadenopathy
Methods line 102: two times “to compensate for this” in one sentence

In summary the study design and rational behind this study are strong, however, the authors could get out significantly more of their own data and hypothesis. Thus, major revision is recommended from my side.

Reviewer 2 ·

Basic reporting

Acceptable

Experimental design

acceptable

no issue here

Validity of the findings

acceptable

Additional comments

In this study , author describes association between IPN (size and number) and COPD severity. Author COPD severity as assessed by GOLD stage classification based on FEV1 correlates with the inflammation which may be associated with enlargement of IPN size and numbers .

Study is well designed and well presented . Figures and tables are clear. However , there is major concern;
-Older GOLD stage classification is based on FEV1 which is not the sole contributor of the COPD severity , 3 years ago new dimensions (rate of exacerbation and dyspnea index ) were added to the new GOLD classification . It will be very useful if authors of this paper has clinical data such as rate of exacerbation , hospitalization or dyspnea score for the correlation. If not then this should be mentioned as a limitation and in addition title should reflect that COPD severity was based on spirometric criteria.

---

## Round 0.2 · Major Revisions

I would like to thank the reviewer for the thoughtful re-reading of this manuscript. Please add a point-by-point reply to the reviewer's comments. I request for the manuscript to be revised accordingly. In addition to these, the statistical methods used in the manuscript should be updated. Please check the normality assumption of data, then use appropriate statistical methods. It may be good to include a biostatistician in the study.

·

Basic reporting

no comment

Experimental design

no comment

Validity of the findings

no comment

Additional comments

Re-Review 4 Dec. 2019

The authors have not really addressed my concerns at all, as they also stated in their rebuttal letter. However, I can accept the manuscript in its present form since it is about an interesting topic and I agree with the authors, that some of my concerns would be too much for this kind of manuscript.
Nevertheless, at least the following points should be clarified:
• A statement from your local ethic committee that the study is fine should be no big problem and would give the official approval to perform your study, which is in the interest of the analyzed patients, you as the authors and the journal you are planning to publish in
• Was a statistician involved in your analyses? It sounds strange that the normality was not tested, this is also a simple test and should be performed anyway before further analyzing the data, there is no good reason for me not to do so
• Why don’t you exclude PRISm from this GOLD 1 & 2 summery and let us know what happens than to the data?
In the end, I leave the decision to the editor. Unfortunately, my reviewers comment were not able to rise the quality of the present manuscript since not a single comment led to a change/improvement.
However, at least the above mentioned major concerns should be addressed in an adequate way.

---

## Round 0.3 · Minor Revisions

The authors have not really addressed the first reviewer's concerns at all. Please provide a detailed answer to convince the reviewer, and address the reviewers’ comments on a point by point basis and make the suggested changes before resubmitting.

·

Basic reporting

no comment

Experimental design

no comment

Validity of the findings

no comment

Additional comments

The authors have in part adequately addressed my issues. From my point of view, I would still give PRISM as own category, so I would prefer the in the rebuttal letter given figure.
EC issue is still not solved. I only know that an amendment to the already approved study should be no big problem. Maybe a call to your IRB would be helpful to ask for a written statement (which is no major issue and should grant to you that everything is fine). Otherwise, I can also accept the granted EC approval for the big study if the editor and Journal do so AND if this is the usual way to handle sub-studies derived from a bigger project in your country. Nevertheless, I know that our IRB would definitely ask for an amendment if a specific side-project is planned out of a bigger study to be published, but this might be different and less strict in another country.
Otherwise, everything is fine to me now.

Reviewer 2 ·

Basic reporting

please see below

Experimental design

please see below

Validity of the findings

please see below

Additional comments

please see below

---

## Round 0.4 · accepted · Accept

The authors addressed the reviewers' concerns and substantially improved the content of MS.

So, based on my own assessment as an editor, no further revisions are required and the MS can be accepted in its current form.

·

Basic reporting

na for this round

Experimental design

na for this round

Validity of the findings

na for this round

Additional comments

for me it is fine now, thank you and all the best